# Effect of Acute and Cumulative Stress on Gene Expression in Mammary Tissue and Their Interactions with Physiological Responses and Milk Yield in Saanen Goats

**DOI:** 10.3390/ani13233740

**Published:** 2023-12-03

**Authors:** Marta Liliane de Vasconcelos, Priscila dos Santos Silva, Henrique Barbosa Hooper, Giovana Krempel Fonseca Merighe, Sandra Aparecida de Oliveira, João Alberto Negrão

**Affiliations:** Department of Basic Sciences, Faculty of Animal Science and Food Engineering, University of Sao Paulo (USP), Pirassununga 13635-900, SP, Brazil; martavasconcelos@usp.br (M.L.d.V.); pri.vet@usp.br (P.d.S.S.); henriquehooper@usp.br (H.B.H.); gkrempel@usp.br (G.K.F.M.); sandoliv@usp.br (S.A.d.O.)

**Keywords:** gene expression, reactive oxygen species, somatic cell count, antioxidants, stress

## Abstract

**Simple Summary:**

Dairy goats face a set of stressful challenges during lactation. In our study, we aimed to understand how daily stressors can impact milk production and also how Saanen goats respond to these stressors. Our results showed that acute stress imposed sequentially caused cumulative stress and decreased the milk yield of a stressed goat. The data obtained may help in milk yields, especially in tropical regions where the use of goats is widely disseminated as a source of income and food. The information from this study can also be extrapolated to research in other production animals.

**Abstract:**

This study addresses the hypothesis that different acute stressors can cumulatively decrease milk yield. In fact, in a time of global warming, the impact of environmental stress and farm management practices on milk production remains unclear. In this context, our objective was to investigate the effect of acute and cumulative stress on gene expression in mammary tissue and their interactions with physiological responses and milk yield in Saanen goats. Thirty lactating goats were subjected to two treatments: (1) control (CT), in which goats were maintained following a habitual routine under comfort conditions; (2) stress (ST), in which the goats were subjected to different types of environmental stress: heat stress, adrenocorticotropic hormone administration, hoof care management, and exposure to rain. These stressors were performed sequentially, with one stress per day on four consecutive lactation days, to evaluate their effect on milk quality and milk yield. Our results showed that compared to CT goats, cumulative stress increased the gene expression of glucocorticoid receptor (GR), interferon-gamma (IFN-γ), superoxide dismutase (SOD), and catalase (CAT) in mammary tissue, which are indicators of cortisol action, inflammatory response, and antioxidant enzymes. Furthermore, the acute challenges imposed on ST goats changed their rectal temperature and respiratory frequency and increased cortisol, glucose, cholesterol, triglycerides, and high-density lipoprotein release in plasma when compared to CT goats. Although these physiological and metabolic responses restore homeostasis, ST goats showed lower milk yield and higher somatic cell count in milk than CT goats. In conclusion, the results confirmed our initial hypothesis that different acute stressors cumulatively decrease the milk yield in Saanen goats.

## 1. Introduction

Dairy goats are subject to a set of stressful farm management practices, and currently, global warming is an additional challenge to copious lactation [1,2,3,4]. However, most studies related to stress in dairy animals imposed a single acute stressor, which is not effective in evaluating how multiple stressors can cumulatively decrease milk yield. Nowadays, heat stress caused by global warming causes a significant decrease in milk production during lactation [5,6,7]. In addition, different farm management practices, such as transport, vaccination, medicine administration, insemination, batch changes, preventive hoof trimming, weaning, and first milking, may cause stress [3,4,8,9]. Although previous studies have shown that each of these different stressors can decrease the milk yield, the effect of cumulative acute stressors on milk yield remains unknown.

There are several environmental and management situations in the routine of goat culture that can trigger acute stress and consequently trigger a series of behavioral and physiological responses to these stressors that lead to a low productive performance [10]. Typically, physiological and behavioral responses allow dairy goats to restore their homeostasis during different stressful challenges [10,11]. However, in dairy animals, stress, via cortisol release, is associated with lower milk yield and higher mastitis risk [12,13]. In the same way, heat stress has been shown to change the expression of target genes related to immunity/inflammation processes and also to have a negative effect on milk yield after the imposed stressors are discontinued [5,12,13]. Moreover, cortisol action in different tissues is affected by the number and responsiveness of glucocorticoid (GR) and mineralocorticoid (MR) receptors in the target tissue [7,9]. However, gene expression and molecular mechanisms associated with cortisol action and immunity/inflammation processes on mammary tissue are not yet fully elucidated. Additionally, stress causes metabolic adjustments that provide extra energy to restore homeostasis [11,14,15].

Indeed, cortisol release causes intense tissue mobilization and metabolic changes during stress that are associated with oxidative processes that increase reactive oxygen species (ROS) and decrease enzymatic and nonenzymatic antioxidants [6,16,17]. Consequently, oxidative stress can cause apoptosis in mammary epithelial cells and decrease milk synthesis [16,17,18]. However, it is not clear how multiple and cumulative stressors can change the gene expression of mammary glands related to the oxidative process, inflammatory responses, and milk synthesis.

For these reasons, we hypothesized that different acute stressors can cumulatively decrease milk yield, changing the expression of target genes related to inflammatory responses and oxidative processes in the mammary tissue. Taking into account four of the most frequent stressors observed on dairy goat farms (heat stress, medicine administration, hoof care, and rain), the objective of this study was to evaluate the effect of different types of acute stress, with one stressor per day, on the expression of target genes related to enzymatic antioxidants of glutathione (GSH), thioredoxin reductase (*TRX*), catalase (CAT), glutathione peroxidase (GPX), superoxide dismutase (SOD), acetyl-CoA carboxylase alpha (ACACA), lipin 1 (LPIN1), inflammatory response interleukins (IL1, IL6, IL8), tumor necrosis factor (TNF-α), and interferon (IFN-γ) in mammary cells, as well as the effect of stress on milk quality and milk yield in Saanen goats.

## 2. Materials and Methods

All animal procedures were approved by the Animal Ethics Committee (protocol 9546150719) of the Faculty of Animal Science and Food Engineering, University of São Paulo (at 21°57′02′′ S, 47°27′50′′ W). This region is classified as subtropical and humid with an average annual rainfall of 1300 to 2000 mm between November and March during the hot and rainy season.

Thirty healthy, nonpregnant, and lactating Saanen goats (means ± sd, body weight of 63.3 ± 11.4 kg, body score of 3.0 ± 0.60, and parity order of 3.0 ± 0.5) were used in this experiment. The experimental goats were raised in a pasture system, housed in collective pens at night, and kept in *Cynodon* spp. (Tifton) paddocks with free access to shade, food, and water troughs during the day. Food was provided twice daily and comprised 50% concentrate (corn grain, soybean meal, soybean oil, and mineral and vitamin mix) and 50% roughage (corn silage) providing 100% of the animals’ nutritional requirements [19]. The goats were milked twice daily (at 06:00 h and 18:00 h), and the milking machine was regulated with a vacuum level of 48 kPa and a pulse rate of 120 cycles/minute.

### 2.1. Experimental Procedures

The goats were randomly distributed into two groups based on parity, previous milk yield, body weight, body score, and number of kids in parturition to the following treatments: stress (ST) or control (CT). The 15 goats in the CT group were maintained following their usual routine under comfort conditions and were not subjected to stressful challenges. The other 15 goats from in the ST group were subjected to 4 stressors sequentially applied: heat stress on day 190 of lactation; ACTH administered on day 191; hoof care on day 192; and rain exposure on day 193. On day 194 of lactation, mammary gland biopsies were performed to study the effect of cumulative stress on gene expression in mammary tissue.

The ST goats were subjected to daily stressors at 10:00 h, while the CT goats were kept under comfort conditions and were not exposed to the different stressors studied. At 15:00 h, the goats subjected to ST and CT treatments were returned to their habitual routine and put in their respective pens or paddocks under similar housing and comfort conditions with access to shade, food, and water following the normal routine of the experimental farm. On the day following the four days of stress imposition, biopsies were performed at 08:00 h after morning milking.

### 2.2. Environmental Conditions and Treatment Imposition

Air temperature (AT) and relative humidity (RH) were measured using a data logger (LogBox-RHT Novus, Miami, FL, USA). The black globe humidity index (BGHI, [20]) was calculated using the following equation: BGHI = [black globe temperature °C) + (0.36 × dewpoint temperature, °C) + 41.5]. The temperature and humidity index (THI, [20]) was calculated using the following equation: THI = [0.8 × air temperature °C)] + [(% relative humidity/100) × (air temperature °C − 14.4)] + 46.4). The AT, RH, THI, and BGHI measured at 08:00 h, 12:00 h, and 18:00 h during the treatment days are shown in Table 1.

On the first experimental day, day 190 of lactation, the ST goats were exposed to the sun for 5 h (between 10:00 h and 15:00 h) in a paddock, without access to shade or water to mitigate the heat stress, and subjected to AT with a mean of 33.5 ± 3.3 °C and RH with a mean of 55.0 ± 5.1%. In this environment, the THI reaches 85 (a value classified as dangerous for goats by Silanikove and Koluman 2015 [21]), and the BGHI reaches 39.7. Simultaneously, the CT goats were maintained in a paddock with free access to shade and water.

On the second experimental day, day 191 of lactation, the ST goats were subjected to 0.6 mg of adrenocorticotropic hormone (ACTH) Kg/PV administered intravenously. This dose was based on previous studies and is considered similar to other acute physiological stressors because cortisol release returns to baseline levels within 6 h after ACTH administration [22]. Simultaneously, the CT goats were subjected to an intravenously administered placebo. The ACTH or placebo was administrated to goats in their habitual paddock, where they had access to shade, water, and total diet. During the challenge (from 10:00 h to 15:00 h), all goats were subjected to AT mean of 26.8 ± 3.3 °C, RH with a mean of 58.4 ± 3.1%, THI of 76, classified as normal for goats by Silanikove and Koluman 2015 [21], and a BGHI of 33.6.

On the third experimental day, day 192 of lactation, the ST goats were subjected to hoof care. This challenge duplicated habitual hoof care procedures on the experimental farm. At 10:00 h, the ST goats were guided one by one to the hoof care box. The hoof trimming procedure lasted from 3 to 10 min per animal and was performed using a hoof knife, scissors, and brush to remove accumulated organic matter and to trim excess hoof. The CT goats were maintained in their habitual paddock with free access to total diet, water, and shade. At 15:00 h, all goats were taken to their habitual pen, where they had access to shade, water, and total diet. During the challenge (10:00 h to 15:00 h), all goats were subjected to an AT mean of 26.6 ± 2.0 °C, an RH mean of 62.7 ± 2.9%, THI reaching 77, classified as normal for goats by Silanikove and Koluman 2015 [21], and BGHI reaching 34.2.

On the fourth experimental day, day 193 of lactation, the ST goats were exposed to rain. As predicted by the National Institute of Meteorology (INMET), this was a warm, cloudy, and rainy day. However, to ensure moderate–strong rainfall in a subtropical area that corresponded to 30 mm of rainfall, the solarium in the habitual pen of the ST goats was equipped with a shading screen and sprinklers to ensure a moderate rain period of 5 h. The shading screen was kept open due to the clouds, and when natural rain stopped for 30 min, the sprinklers were used to guarantee between 5 and 10 mm of water per hour. The CT goats were maintained in their habitual pens (with covered area and solarium) because farm goats are routinely kept in their pens during cloudy and rainy days. During the challenge (10:00 h to 15:00 h), the goats were subjected to an air temperature of 29.2 ± 2.5 °C, relative air humidity of 72.0 ± 2.3% (for rain-exposed goats, the RH was close to 100%), THI reaching 76 (classified as normal for goats [21]), and BGHI reaching 34.5.

### 2.3. Physiological Data and Milk and Blood Sampling and Analyses

On the first, second, third, and fourth experimental days (lactation days 190–194), heart rate (HR), respiratory rate (RR), rectal temperature (RT), and blood samples were evaluated before, during, and after imposition of the ST or CT treatments, at 09:30 h, 12:30 h, and 15:30 h, respectively. HR was measured in the thoracic region using a stethoscope for 1 min. RR was measured by visual observation of the goats’ flank movements for 1 min. RT was measured with a thermometer; afterward, blood samples were taken from the jugular veins of the experimental goats.

Blood samples from the jugular of goats were also collected at 09:30 h (before), 13:00 h (during), and 16:30 h (after) the treatments for the ST and CT group; the samples were centrifuged at 3000 rpm for 17 min at 4 °C, and plasma obtained was stored at –20 °C. Plasmatic cortisol (CORT) concentration was measured using an enzyme immunoassay kit (Cortisol TestSystem, Monobind, Lake Forest, CA, USA). Metabolite concentrations were measured using enzymatic kits for glucose (Glicose Laborlab, Guarulhos, SP, Brazil), triglycerides (Triglicerideos Laborlab, Guarulhos, SP, Brazil), cholesterol (Colesterol Total, Laborlab, Guarulhos, SP, Brazil), urea (Urea, Bioclin, Belo Horizonte, MG, Brazil), total protein (Proteina total, Bioclin, Belo Horizonte, MG, Brazil), high-density lipoprotein (HDL, Laborlab, Guarulhos, SP, Brazil), and low-density lipoprotein (LDL, Laborlab, Guarulhos, SP, Brazil).

Individual milk production was measured daily. Milk samples were collected weekly from both teats of each goat for microbiological analysis. A representative portion of the total milk of each goat was also sampled weekly to determine its fat, protein, lactose, and somatic cell count (SCC). The fat, protein, and lactose content in the milk was measured using ultrasound equipment (MilkScope Expert Razgrad, Bulgaria). The milk’s somatic cell count (SCC) was performed in smears (10 µL) stained with pyronin Y and methyl green, and somatic cells were counted using the direct microscopy method [23]. Microbiological analysis of the milk was performed using the total colony counts method [24]. Baird-Parker agar was used for the identification of *Staphylococcus* sp., and MacConkey agar was used for the identification of *Enterobacteriaceae*. After spreading the milk sample, the microbiological culture was incubated for 48 h at 37 °C. The results of the microbiological analysis were presented as colony-forming units per milliliter of milk (CFU mL^−1^).

### 2.4. Biopsy and Gene Expression

Biopsies were performed after morning milking (at 08:00 h) on the day after stressor impositions (on lactation day 194). As experimental goats presented homogeneous parity, body weight, body score, and milk yield, representative mammary tissue from 14 goats (7 samples from CT and 7 from ST goats) was collected to study the effect of cumulative stress. Biopsy procedures and animal care have been described previously [7,9].

The expression of target genes was determined with RT-PCR equipment (Invitrogen, 169 Burlington, ON, Canada), using the primers of glucocorticoids receptor (GR), glutathione (GSH), thioredoxin reductase (TRX), catalase (CAT), glutathione peroxidase (GPX), superoxide dismutase (SOD), acetyl-CoA carboxylase alpha (ACACA), lipin 1 (LPIN1), tumor necrosis factor (TNF-α), interferon (IFN-γ), and interleukins 1, 6, and 8 (IL-1, IL-6, and IL-8), as described in Table 2.

The RNA from these samples was extracted and purified using a PureLink RNA Mini Kit (Invitrogen, Burlington, ON, Canada). RNA concentrations were determined using Qubit 2.0 (Thermo Fisher Scientific, Waltham, MA, USA). The total RNA obtained was treated with RNase-free Dnase (Promega, Madison, WI, USA). The RNA was transcribed to cDNA using the GoScript Reverse Transcriptase kit (Promega, Madison, WI, USA). Genes were amplified, and each reaction was performed in duplicate. In the present study, phospholipase A2 (*YWHAZ*), glyceraldehyde-3-phosphate dehydrogenase (*GAPDH*), and ubiquitin C (*UBC*) gene expressions were used as reference, and the expression of target genes was calculated using Livak’s method [25].

### 2.5. Statistical Analysis

The data were analyzed using the Statistical Analysis System (SAS Intitute Inc., Car, NC, USA, 2008). The normality of the data was confirmed using Shapiro–Wilk’s test. Heart rate (HR), respiratory rate (RR), rectal temperature (RT), cortisol (CORT), and metabolites were subjected to an analysis of variance following the MIXED procedure. In this analysis, treatment (ST or CT) and time of sampling were considered fixed effects, and goats were considered random effects. Milk yield, milk composition, and somatic cell count were subjected to analysis of variance using the MIXED procedure. In this analysis, treatment (ST or CT) and day of lactation were considered fixed effects, and goats were considered random effects. Gene expression in mammary tissue was subjected to an analysis of variance via GLM procedure, which considered treatment (ST or CT) as a fixed effect and goats as random effects. Several errors of covariance structures were tested, and the one that best fit the data according to the Bayesian information criterion was selected. The means were compared using Fisher’s exact test when there was a significant effect. Statistical significance was defined as *p* ≤ 0.05.

## 3. Results

### 3.1. Physiological Data, Cortisol, and Metabolites

There was significant interaction between treatment and time of sampling on cortisol (CORT) release in ST goats when compared to CT goats. There was a significant increase in CORT concentration during all stressors, and CORT returned to baseline concentrations after ACTH, hoof care, and rain stressors. In goats subjected to heat stress, however, the CORT concentration remained significantly high when compared to the CT goats (Figure 1).

There was also significant interaction between treatment and time of measurement on respiratory rate (RR) and rectal temperature (RT); however, there was no effect of stress imposed on heart rate (HR) (Table 3). In fact, RR observed during heat stress was significantly higher in ST goats when compared to CT goats (96.80 ± 6.15 versus 78.30 ± 4.60 at 12:30 h). However, RR observed after rain stress was significantly lower for ST goats than CT goats (40.03 ± 2.40 versus 73.25 ± 5.72 at 15:30 h). Furthermore, RT measured during heat stress was significantly higher in ST goats when compared to CT goats (39.47 ± 0.07 versus 38.65 ± 0.05 at 12:30 h). Similarly, RT measured during hoof care stress was significantly higher in ST goats than in CT goats (39.47 ± 0.07 versus 38.65 ± 0.05 at 12:30 h).

The treatments had a significant effect on glucose (GLUC), cholesterol (CHOL), triglycerides (TG), and high-density lipoprotein (HDL), and ST goats showed significantly higher GLU, CHOL, TG, and HDL when compared to CT goats. However, the treatments had no significant effect on low-density lipoprotein (LDL), urea (UREA), and total protein (TP) (Table 4).

### 3.2. Milk Yield and Quality

There was interaction between treatment within lactation day and milk yield; in fact, during (on lactation days 192 and 193) and after (on lactation days 197 and 198) imposition of treatments, ST goats showed significantly less milk yield compared to CT goats (Figure 2). However, there was no treatment effect on fat, protein, and lactose percentages or on *Staphylococcus* and *Enterobacteriaceae* count in milk (Table 5). There was significant effect of stressors on somatic cell count (SCC) in milk, and ST goats showed a higher SCC mean when compared to CT goats (Table 5).

### 3.3. Gene Expression

Taking into account the gene expressions in mammary tissue, ST goats showed higher expression of GR, SOD, CAT, and IFN-y genes when compared to CT goats. However, there was no significant effect of treatment on expression of GSH, TRX, GPX, ACACA, IL-1, IL-6, IL-8, and TNF-α genes in the mammary tissue of Saanen goats (Table 6).

## 4. Discussion

The different stressors imposed in this study were considered acute and cumulative stressors. In fact, all stressors imposed sequentially, one per day, significantly affected CORT, RR, and RT, with a result of restoring the homeostasis of ST goats. In the present study, higher CORT and RR were associated with physiological mechanisms to maintain RT in response to different stressors imposed. During heat stress, ST goats showed higher RR, a mechanism for lowering body temperature through breathing, while during rain stress, ST goats showed significantly lower RR, which can maintain RT. Some previous studies reported similar changes when air temperature, relative humidity, and THI values change across sunny and rainy days in tropical and subtropical regions [26,27,28]. Other studies reported similar results and suggested that these same physiological responses showed that stressed goats cope adequately with different stressors to restore their homeostasis some hours after the end of stressor imposition [9,15]. However, in our study, the cumulative stress caused by different acute stressors decreased milk yield, confirming our initial hypothesis.

Taking into account the CORT release, the four acute stressors studied sequentially here had different intensities and demonstrated the adrenal responsiveness of Saanen goats during lactation. Although ACTH administration showed high CORT concentration when compared to heat stress, hoof care stress, and rain stress, the CORT profile observed in the present study was considered similar to other acute stressors related to farm environment and management practices [9,29,30]. In addition, the cumulative stress imposed in our study also increased the expression of the GR gene in mammary tissue when compared to CT goats. As GR is the main CORT receptor, other studies argued that stress imposition and higher expression of the GR gene improved CORT´s responsiveness in mammary tissue [9,31]. Indeed, some studies demonstrated that higher CORT release and higher expression of GR in mammary tissue were associated with a higher apoptosis rate and decrease in milk synthesis [7,9]. In this context, it is possible to argue that acute stressors cumulatively decrease milk synthesis via apoptosis of mammary cells. However, further studies are necessary to understand the interaction among cumulative stress, mammary cell apoptosis, and milk yield.

In addition, other authors argued that stress, via CORT release, promotes metabolic adjustments to provide the energy necessary to support the physiological responses to return to homeostasis [14,15,27]. In the present study, CORT increase was related to a significant increase in GLUC, CHOL, HDL, and TG in ST goats. As previously reported [14,32], it is possible to argue that stressors imposed in the present study induced gluconeogenesis and lipolysis in ST goats when compared to CT goats. In this way, our results suggest that the stressors imposed caused metabolic adjustments, providing extra energy for ST goats to maintain their homeostasis. Furthermore, stress is also associated with the accumulation of reactive oxygen species (ROS) caused by an imbalance in the synthesis of oxidant and antioxidant molecules [17,18,33]. Indeed, the increase in ROS via stress was related to cell apoptosis and tissue damage [10,18,33]. In contrast, in our study, ST goats showed a higher expression of antioxidant genes SOD and CAT when compared to CT goats, suggesting that mammary tissue increases antioxidant enzyme synthesis as a protective response to mitigate the negative effects caused by cumulative stress. In this context, further studies are necessary to understand how cumulative stressors affect the synthesis of oxidant and antioxidant molecules in mammary tissue.

Furthermore, it is well established that CORT concentration is the main biomarker of stress in livestock animals, and a cortisol profile is used to establish acute and chronic stress. In fact, during acute stress, cortisol release takes place over minutes and hours, while during chronic stress, cortisol release takes place over days or weeks as a way to return to baseline concentrations. Consequently, acute stress and chronic stress have different impacts on animal production [9,10,11]. Various authors reported that acute stress did not change milk yield, but chronic stress is mostly related to decreased milk yield [5,9,32]. In our study, although the physiological and metabolic responses caused by different acute stressors restored homeostasis, the ST goats showed lower milk yield than the CT goats. In our study, the lower milk yield of the ST goats was not caused by mastitis because the experimental goats did not present clinical or subclinical symptoms of mastitis, and total counts of *Enterobacteriaceae* and *Staphylococcus* sp. in their milk showed adequate status and were also similar to those measured in milk from healthy goats [34,35].

In our study, cumulative stress increased the gene expression of IFN-y in ST goats, indicating an inflammatory response in mammary tissue when compared with CT goats. Although ST goats showed a significant increase in SCC in their milk when compared to CT goats, there was no change in the bacterial count in milk and no increase in clinical and subclinical mastitis rates in ST goats. Consequently, we cannot argue that cumulative acute stress imposed sequentially changes the immune response in Saanen goats. In fact, our results were similar to those described previously in which the researchers did not observe any differences in SCC or mastitis risk when different acute stressors were imposed on Saanen goats [7,9,28]. Furthermore, other studies argued that higher SCC in the milk of healthy goats is a consequence of cell apoptosis and greater exfoliation of mammary epithelial cells [36,37]. In contrast, other studies of acute stressors demonstrated that higher cortisol concentration was related to an increase in SCC in milk and interleukins in plasma, suggesting that stress increased the mastitis risk in dairy ewes [12,13]. These contrasting results show that further studies are necessary to understand the interactions among stress, SCC, and immune–inflammatory processes in mammary glands.

## 5. Conclusions

Saanen goats subjected to cumulative stress upregulated the expression of glucocorticoid receptor (GR), IFN-γ (inflammatory response), and SOD and CAT (antioxidant enzymes) genes in mammary tissue and produced increased somatic cell counts (SCCs) in their milk. Furthermore, cumulative stress triggered physiological responses (CORT, RR, and RT) and induced metabolic adjustment (GLUC, CHOL, and HDL) to help the goats return to homeostasis. Finally, cumulative stress decreased milk yield on Saanen goats, confirming our initial hypothesis.

## Figures and Tables

**Figure 1 animals-13-03740-f001:**
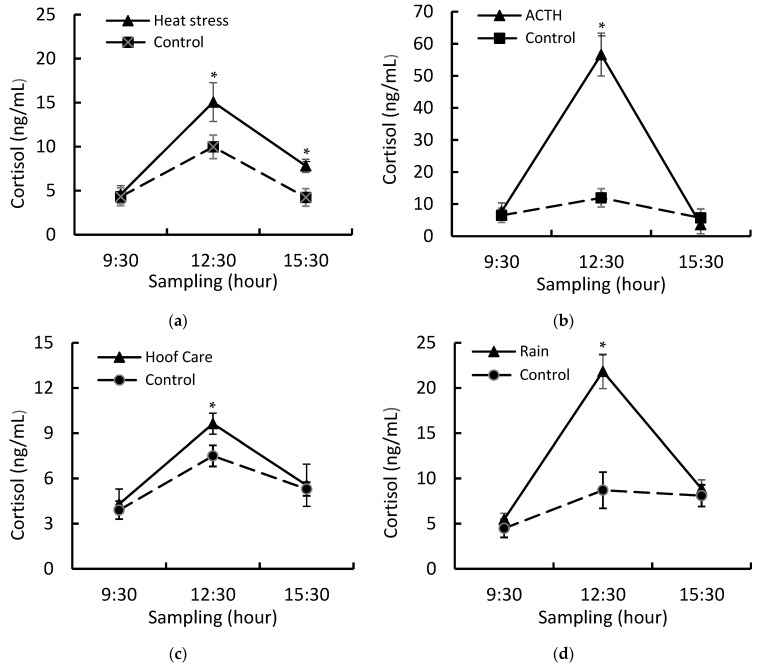
Cortisol concentrations in Saanen goats before (09:30 h), during (12:30 h), and after (15:30 h) imposition of stress (ST) or control treatments (CT): (**a**) goats subjected to heat stress or control on day 190 of lactation; (**b**) goats subjected to ACTH or control on day 191 of lactation; (**c**) goats subjected to hoof care or control on day 192 of lactation; (**d**) goats subjected to rain or control on day 193 of lactation. Data are presented as mean ± standard error of means (SEM). Means with * differ (*p* ≤ 0.05).

**Figure 2 animals-13-03740-f002:**
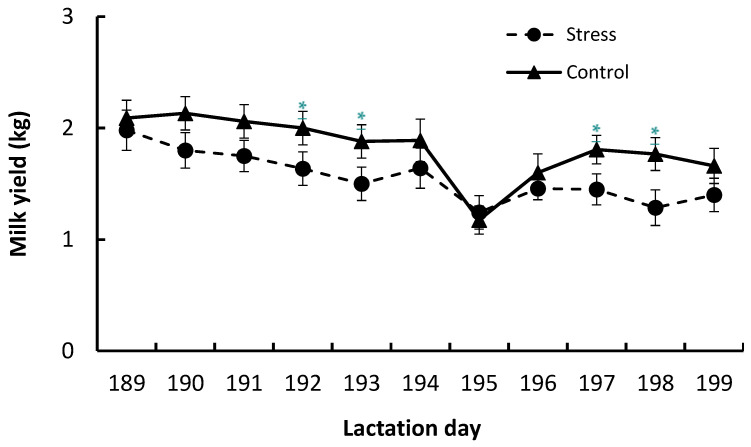
Milk yield of Saanen goats subjected to stress (ST) or control (CT) treatment before, during (lactation days 190 to 194), and after stress challenges. On lactation day 190, goats were subjected to heat stress or control treatment; on day 191 they were subjected to ACTH administration or control treatment; on day 192 they were subjected to hoof care or control treatment; and on day 193 they were subjected to rain or control treatment. Data are presented as mean ± standard error of mean (SEM). Means with * differ (*p* ≤ 0.05).

**Table 1 animals-13-03740-t001:** Air temperature (AT), relative humidity (RH), temperature humidity index (THI), and black globe temperature and humidity index (BGHI) measured at 08:00 h, 12:00 h, and 18:00 h during the treatment days.

Trait		08:00 h	12:00 h	18:00 h
1st day 190—lactation	AT *	26.8	33.0	26.3
	RH *	71.2	54.9	51.6
	THI	77	83	74
	BGHI	31.8	39.2	31.3
2nd day 191—lactation	AT	26.0	28.0	25.5
	RH	54.1	59.9	66.2
	THI	76	74	74
	BGHI	33.3	30.9	30.3
3rd day 192—lactation	AT	26.8	27.5	24.4
	RH	67.5	64.5	59.1
	THI	77	76	72
	BGHI	32.7	31.8	29.0
4th day 193—lactation	AT	25.5	27.9	28.5
	RH	63.1	65.2	68.2
	THI	74	78	79
	BGHI	30.3	33.1	33.9

* AT (°C); RH (%).

**Table 2 animals-13-03740-t002:** Primer sequences used in the experiment.

Gene ^1^	Primer Sequences	GenBank Code	Amplicon
*GR*	3′ CCATTTCTGTTCACGGTGTG 5′	XM_005683087	132
	5′ CTGAACCGACAGGAATTGGT 3′		
*SOD*	3′ TGTTGCCATCGTGGATATTGTAG 5′	NM_001285550	102
	5′ CCCAAGTCATCTGGTTTTTCATG 3′		
*GPX*	3′ GCAAGGTGCTGCTCATTGAG 5′	XM_005695962	82
	5′ CGCTGCAGGTCATTCATCTG 3′		
*CAT*	3′3′ GCTCCAAATTACTACCCCAATAGC 5′	NM_001035386	104
	5′ GCACTGTTGAAGCGCTGTACA 3′		
*GSH*	3′3′ GAGAACGCTGGCATTGAG 5′	NM_001114190	143
	5′ AGCAGGCAGTCAACATCT 3′		
*TRX*	3′ AGGAGAAAGCTGTGGAGAAA 5′	NM_174625	94
	5′ TTATCCCTTGATGGAATCGT 3′		
*TNF-α*	3′ TCTTCTCAAGCCTCAAGTAACAAGC 5′	EU276079	103
	5′ CCATGAGGGCATTGGCATAC 3′		
*IL-1β*	3′ TCCACCTCCTCTCACAGGAAA 5′	NM_174093	99
	5′ TACCCAAGGCCACAGGAA 3′		
*IL6*	3′ TGCTGGTCTTCTGGAGTATC 5′	NM_173923	153
	5′ GTGGCTGGAGTGGTTATTAG 3′		
*IL8*	3′ TGTGAAGCTGCAGTTCTGTCAA 5′	AF232704	130
	5´ TTTCACAGTGTGGCCCACTCT 3′		
*IFN-γ*	3′ TTCAGAGCCAAATTGTCTCC 5′	NM_174086	205
	5′ AGTTCATTTATGGCTTTGCGC 3′		
*ACACA*	3′ TGGTCTGGCCTTACACATGA 5′	NM_174224	112
	5′ TGCTGGAGAGGCTACAGTGA 3′		
*LPIN1*	3′ GAGGGGAAGAAACACCACAA 5′	NM_001206156	202
	5′ GTAGCTGACGCTGGACAACA 3′		
*GAPDH*	3′ GGTGATGCTGGTGCTGAG 5′	NM_001034034	181
	5′ TGACAATCTTGAGGGTGTTG 3′		

^1^ Glutathione (GSH), thioredoxin reductase (TRX), catalase (CAT), glutathione peroxidase (GPX), superoxide dismutase (SOD), acetyl-CoA carboxylase alpha (ACACA), lipin 1 (_LPIN1_), interleukins 1, 6, and 8 (IL-1, IL-6, and IL-8), glyceraldehyde-3-phosphate dehydrogenase (GAPDH), tumor necrosis factor (TNF-α), and interferon (IFN-γ).

**Table 3 animals-13-03740-t003:** Rectal temperature, respiratory rate, heat rate, and cortisol of Saanen goats subjected to treatments. Data correspond to mean measured on day 190 of lactation of goats subjected to heat stress or control treatment; on day 191 of lactation of goats subjected to ACTH administration or control treatment; on day 192 of lactation of goats subjected to hoof care or control treatment; and on day 193 of lactation of goats subjected to rain or control treatment. Data are presented as means ± standard error of mean.

**Physiological Data ^1^**	**Heat Stress**	**Control**	** *p* ** **-Value**		
			**T ^2^**	**H ^2^**	**T*H ^2^**
RT	39.20 ± 0.10	38.97 ± 0.05	0.35	0.01	0.01
RR	78.34 ± 3.90	74.41 ± 3.71	0.04	0.01	0.01
HR	100.51 ± 1.30	98.95 ± 2.34	0.25	0.11	0.23
**Physiological Data ^1^**	**ACTH**	**Control**	** *p* ** **-Value**		
			**T**	**H**	**T*H**
RT	38.91 ± 0.08	38.95 ± 0.06	0.15	0.01	0.10
RR	52.50 ± 4.53	54.29 ± 3.72	0.40	0.01	0.21
HR	98.04 ± 1.93	98.29 ± 1.77	0.23	0.01	0.16
**Physiological Data ^1^**	**Hoof Care**	**Control**	** *p* ** **-Value**		
			**T**	**H**	**T*H**
RT	39.09 ± 0.05	39.02 ± 0.06	0.38	0.01	0.01
RR	36.20 ± 2.64	38.90 ± 2.89	0.45	0.01	0.20
HR	99.64 ± 1.56	101.10 ± 2.00	0.48	0.03	0.37
**Physiological Data ^1^**	**Rain**	**Control**	** *p* ** **-Value**		
			**T**	**H**	**T*H**
RT	38.98 ± 0.07	38.52 ± 0.05	0.26	0.01	0.01
RR	32.45 ± 2.40	49.40 ± 3.62	0.33	0.01	0.01
HR	100.62 ± 1.84	98.67 ± 1.16	0.22	0.01	0.12

^1^ Rectal temperature (RT), respiratory rate (RR), heat rate (HR), cortisol (CORT). ^2^ T—treatment, H—hours, treatment/hour (T*H) effect.

**Table 4 animals-13-03740-t004:** Glucose (GLU), urea, total protein (TP), triglycerides (TG), cholesterol (CHOL), high-density lipoprotein (HDL), and low-density lipoprotein (LDL) in plasma of goats subjected to stress (ST) or control (CT) treatments. Data correspond to mean measured from 190 to 194 days of lactation for ST and CT goats. Data are presented as means ± standard error of mean.

Measurements ^1^ Mg/dL	Stress	Control	*p*-Value		
			T ^2^	H ^2^	T*H ^2^
GLU	75.80 ^a^ ± 3.01	72.40 ^b^ ± 3.77	0.04	0.01	0.09
UREA	55.25 ± 2.13	56.33 ± 2.15	0.10	0.01	0.01
TP	74.56 ± 1.86	73.90 ± 1.75	0.23	0.32	0.40
TG	18.27 ^a^ ± 0.32	15.88 ^b^ ± 0.46	0.01	0.07	0.62
CHOL	106.42 ^a^ ± 2.74	99.29 ^b^ ± 3.83	0.01	0.02	0.06
HDL	32.45 ^a^ ± 0.51	30.87 ^b^ ± 0.51	0.02	0.40	0.73
LDL	64.35 ± 4.19	69.78 ± 4.73	0.15	0.01	0.10

^1^ Glucose (GLU), urea, total protein (TP), triglycerides (TG), cholesterol (CHOL), high-density lipoprotein (HDL), low-density lipoprotein (LDL). ^2^ T—treatment, H—hours, treatment/hour (T*H) effect. ^a,b^ Means within a row with different superscripts differ (*p* ≤ 0.05).

**Table 5 animals-13-03740-t005:** Milk yield, fat, protein, and lactose percentage. Somatic cell count (SCC) and total counts of *Enterobacteriaceae* and *Staphylococcus* sp. in milk of goats subjected to stress (ST) or control (CT) treatments. Data concerning milk yield. Microbiological analysis of milk composition corresponds to mean measured from lactation days 190 to 194 for ST and CT goats. Data are presented as means ± standard error of mean.

Milk Yield and Milk Quality ^2^	Stress	Control	*p*-Value		
			T ^1^	H ^1^	T*H ^1^
Milk yield (kg)	1.76 ± 0.15	2.09 ± 0.23	0.10	0.18	0.04
Fat (%)	4.25 ± 0.40	4.60 ± 0.20	0.31	0.69	0.75
Protein (%)	3.35 ± 0.36	3.40 ± 0.25	0.82	0.15	0.11
Lactose (%)	4.04 ± 0.66	4.03 ± 0.60	0.81	0.14	0.13
SCC (cells/mL)	1 660 ^a^ ± 203	1 215 ^b^ ± 119	0.05	0.15	0.01
Total bacterial count (CFU/mL)	20.50 ± 10.86	14.26 ± 12.31	0.57	0.62	0.50
Enterobacteriaceae (CFU/mL)					
*Staphylococcus* sp. (CFU/mL)	10.4 ± 3.4	9.35 ± 4.40	0.21	0.25	0.33

^1^ Treatment (T), hour (H), treatment/hour (T*H) effect. ^2^ Somatic cell count (SCC) values × 10^3^/mL, colony-forming units count (CFU), insufficient data for statistical analysis. ^a,b^ Means within a row with different superscripts differ (*p* ≤ 0.05).

**Table 6 animals-13-03740-t006:** Gene expression (mRNA: 2^−ΔΔCT^ method) of target genes in the mammary tissue of goats subjected to ST or CT treatments. To evaluate the effects of treatments (ST or CT) in experiment on gene expression, we calculated the fold change relative to the placebo goats (∆∆Ct) as shown: ∆∆Ct = 2 − (∆Ct ST goats − ∆Ct CT goats). Mammary biopsies were obtained at 194 days of lactation. Data are presented as means ± standard error of mean.

Gene Expression ^1^	Stress	Control	*p*-Value T ^2^
GR	1.34 ^a^ ± 0.09	1.02 ^b^ ± 0.02	0.03
SOD	1.70 ^a^ ± 0.11	1.14 ^b^ ± 0.06	0.01
GSH	0.99 ± 0.14	2.77 ± 1.41	0.39
GPX	1.07 ± 0.31	1.00 ± 0.03	0.38
CAT	1.33 ^a^ ± 0.08	1.00 ^b^ ± 0.04	0.02
TRX	1.09 ± 0.12	1.33 ± 0.26	0.57
IFN-γ	2.89 ^a^ ± 0.28	1.34 ^b^ ± 0.24	0.01
TNF-a	2.21 ± 0.45	1.33 ± 0.27	0.25
IL-1	1.38 ± 0.16	1.21 ± 0.26	0.69
IL-6	4.14 ± 0.82	1.63 ± 0.48	0.17
IL-8	3.01 ± 0.50	2.76 ± 1.29	0.89
LPIN1	1.67 ± 0.33	1.50 ± 0.25	0.77
ACACA	1.62 ± 0.23	1.09 ± 0.25	0.22

^1^ Glucocorticoid receptor (GR), superoxide dismutase (SOD), glutathione (GSH), glutathione peroxidase (GPX), catalase (CAT), thioredoxin reductase (TRX), enzyme acetyl-CoA carboxylase alpha (ACACA), protein lipin 1 (LPIN1), interferon (IFN-γ), tumor necrosis factor (TNF-a), and interleukins 1, 6, and 8 (IL-1, IL-6, and IL-8). ^2^ Treatment (T). ^a,b^ Means within a row with different superscripts differ (*p* ≤ 0.05).

## Data Availability

The reported results can be found at: https://doi.org/10.11606/D.74.2021.tde-30042021-160931. Accessed on 13 November 2023.

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
