# Peer review of "Effect of Acute and Cumulative Stress on Gene Expression in Mammary Tissue and Their Interactions with Physiological Responses and Milk Yield in Saanen Goats"

_animals, 2023, doi:10.3390/ani13233740_

Round 1

Reviewer 1 Report

Comments and Suggestions for Authors

Comments

The authors set out to test the cumulative effect of multiple stressors on milk yield and other physiological parameters such as heart rate, respiratory rate, and body temperature in dairy goats whose milk demand is on the rise because of better digestibility and certain therapeutic advantages compared with milk from other animals [1–3]. Dairy goats, especially Saanen, tend to be affected by environmental stressors such as wet conditions. Studies that investigate these stressors should be interesting. Although the authors show the effect of multiple stressors, comparing this with individual stressors ought to have been part of the study. This would answer the question as to whether certain stressors infer greater stresses than others. However, since they managed to collect samples each day they performed the study, they should capitalize on this approach. Therefore, Tables 4 and 5 should be turned into figures to be able to show, to some level, the effect of the stresses on each day. Computation using a repeated measures analysis (if they sampled the same goats) should show the difference between treatments on the specific days of the stressors. Also, they should more explicitly compare their combined multiple stressors with other studies that looked at individual stresses.

Regarding the writing, the manuscript needs thorough English language improvement and copy editing in all sections. The sections for results and especially discussion need thorough revision and editing. It was so hard to comprehend them.

Specific comments

Line 28: Improve sentence presentation.

Line 28, 389: You talk a lot about somatic cell count. However, some researchers argue that SCC is not very important in dairy goats. I think you can report it, however, minimize making conclusions on it. I think you can focus on drawing conclusions on other interesting physiological outcomes, such as body temperature.

Line 33-40: Remove

Line 66-67: Improve sentence presentation.

Line 73-75: Instead of just listing everything, can you categorize these gene markers such as inflammatory markers, oxidative stress, etc.

Line 118: Sentence repeated as in 104

Line 146: Via what?

Line 288: Looks like the * is missing on some of the figures

Line 324: The interpretation of the interaction and subsequent conclusion seems a bit inaccurate. I am predicting that on certain days the milk yield was significantly higher but lower on other days resulting in a significant interaction. A figure and post hoc analyses will be able to show this more explicitly.

Line 364: I suppose should be “gluconeogenesis”

Line 389-395: The statements seem vague. Need an improvement in the logical flow of the ideas.

Line 402: How can an increase in SCC cause an increase in SCC?

References

1.          Claeys, W.L.; Verraes, C.; Cardoen, S.; De Block, J.; Huyghebaert, A.; Raes, K.; Dewettinck, K.; Herman, L. Consumption of Raw or Heated Milk from Different Species: An Evaluation of the Nutritional and Potential Health Benefits. Food Control 2014, 42, 188–201, doi:10.1016/j.foodcont.2014.01.045.

2.          Park, Y.W.; Juárez, M.; Ramos, M.; Haenlein, G.F.W. Physico-Chemical Characteristics of Goat and Sheep Milk. Small Ruminant Research 2007, 68, 88–113, doi:10.1016/j.smallrumres.2006.09.013.

3.            Parmar, H.; Hati, S.; Sakure, A. In Vitro and In Silico Analysis of Novel ACE-Inhibitory Bioactive Peptides Derived from Fermented Goat Milk. Int J Pept Res Ther 2018, 24, 441–453, doi:10.1007/s10989-017-9630-4.

Comments on the Quality of English Language

The manuscript requires a thorough improvement in Scientific writing including improvements in logical flow and copy editing.

Author Response

Reviewer 1, general comments

The authors set out to test the cumulative effect of multiple stressors on milk yield and other physiological parameters such as heart rate, respiratory rate, and body temperature in dairy goats whose milk demand is on the rise because of better digestibility and certain therapeutic advantages compared with milk from other animals [1–3]. Dairy goats, especially Saanen, tend to be affected by environmental stressors such as wet conditions. Studies that investigate these stressors should be interesting. Although the authors show the effect of multiple stressors, comparing this with individual stressors ought to have been part of the study. This would answer the question as to whether certain stressors infer greater stresses than others. However, since they managed to collect samples each day they performed the study, they should capitalize on this approach. Therefore, Tables 4 and 5 should be turned into figures to be able to show, to some level, the effect of the stresses on each day. Computation using a repeated measures analysis (if they sampled the same goats) should show the difference between treatments on the specific days of the stressors. Also, they should more explicitly compare their combined multiple stressors with other studies that looked at individual stresses.

Regarding the writing, the manuscript needs thorough English language improvement and copy editing in all sections. The sections for results and especially discussion need thorough revision and editing. It was so hard to comprehend them.

Author response: Thank you for your suggestions. We understand the reviewer’s concerns and have included Figure 2 to show how the milk yield was affected by each stressor. At the same time, we improved the text of the Results section to clarify the treatment effect within each stressor. For example, lines 233-238 deal with cortisol and lines 245-254 with physiology in the new version of the manuscript. However, there was no significant interaction between metabolite data and milk content, and for this reason we retain Tables 4 and 5. 

Specific comments

Line 28: Improve sentence presentation.

Authors: reformulated.

Line 28, 389: You talk a lot about somatic cell count. However, some researchers argue that SCC is not very important in dairy goats. I think you can report it, however, minimize making conclusions on it. I think you can focus on drawing conclusions on other interesting physiological outcomes, such as body temperature.

Line 389: In the present study the higher SCC in milk and expression of IFN-y gene on ST goats was related to increase of SCC in milk and consequently associated to greater cell apoptosis. In our study, the increase of expression IFN-y gene in mammary tissue was related to inflammatory responses, because there was no change bacterial count in milk neither increase of mastitis rate on ST goats.

Authors: reformulated (line 28 and 1157).

Line 33-40: Remove

Authors: removed

Line 66-67: Improve sentence presentation.

Authors: The whole introduction has been reworked.

Line 73-75: Instead of just listing everything, can you categorize these gene markers such as inflammatory markers, oxidative stress, etc.

Authors: The whole introduction has been reworked.

Line 118: Sentence repeated as in 104.

Authors: Line 118 was excluded.

Line 146: Via what?

Authors: ACTH administration was added.

Line 288: Looks like the * is missing on some of the figures

Authors: We have checked all figures, and when appropriate added * on respective mean.

Line 324: The interpretation of the interaction and subsequent conclusion seems a bit inaccurate. I am predicting that on certain days the milk yield was significantly higher but lower on other days resulting in a significant interaction. A figure and post hoc analyses will be able to show this more explicitly.

Authors: We added Figure 2 to the manuscript concerning the milk yield and improved the text of the Results section to make clear the cumulative effect different acute stressors imposed on milk yield.

Line 364: I suppose should be “gluconeogenesis”

Authors: Yes, the term was added as suggested

Line 389-395: The statements seem vague. Need an improvement in the logical flow of the ideas.

Authors: We agree that physiological responses need to be highlighted. In addition, we improved our discussion concerning the SCC in milk. These points were added in lines 312-324 and lines 369-386 of the new version of the manuscript.

Line 402: How can an increase in SCC cause an increase in SCC?

Authors: This sentence was improved as follows: other studies have argued that higher SCC in the milk of healthy goats is a consequence of cell apoptosis and greater exfoliation of mammary epithelial cells (lines 377-379 in the new version).

Reviewer 2 Report

Comments and Suggestions for Authors

This study, examining the effects of various acute stressors on Saanen goats, displays some notable strengths but also has areas that require improvement.

  1. Post Hoc Test Not Specified: The study does not mention which specific post hoc test was employed. This is a critical aspect, as the choice of test can influence the interpretation of results.
  2. Vague Conclusion: The conclusion lacks specificity and does not adequately summarize the findings. It should provide clear insights into the observed effects of stressors on metabolic adjustments, gene expression, and milk yield.
  3. Limited Use of References: The paper could benefit from a more comprehensive review of relevant literature. Including more references would provide a stronger theoretical framework for the study.
  4. Lack of References in Discussion: Several statements in the discussion section lack proper referencing, making it challenging for readers to verify or contextualize the claims made.
  5. Novelty and Hypothesis: The study does not clearly articulate its novelty in relation to existing literature. It's crucial to emphasize what sets this study apart from similar research. Additionally, the statement about successfully solving the hypothesis needs more robust evidence or explanation.
  6. Ambiguity in Study Design and Methods: The study's methodology could be more detailed. For example, the specifics of the administration of adrenocorticotropic hormone (ACTH) are not mentioned, which is crucial for replication and understanding of the results.

Overall, while the study offers valuable insights into the effects of acute stressors on Saanen goats, addressing the above points would significantly strengthen its scientific rigor, clarity, and contribution to the field.

Top of Form

Author Response

Reviewer 2, general comments

This study, examining the effects of various acute stressors on Saanen goats, displays some notable strengths but also has areas that require improvement.

Authors: The entire manuscript was revised to improve English.

Post Hoc Test Not Specified: The study does not mention which specific post hoc test was employed. This is a critical aspect, as the choice of test can influence the interpretation of results.

Authors: We have improved the information concerning the post hoc test in the statistical analysis section (lines 217-230).

Vague Conclusion: The conclusion lacks specificity and does not adequately summarize the findings. It should provide clear insights into the observed effects of stressors on metabolic adjustments, gene expression, and milk yield.

Authors: The conclusion was rewritten to summarize the main physiological and metabolic results and to report the effects of cumulative stress on gene expression and milk yield, confirming our initial hypothesis (lines 384-391).

Limited Use of References: The paper could benefit from a more comprehensive review of relevant literature. Including more references would provide a stronger theoretical framework for the study.

Authors: We have rewritten the Introduction and Discussion sections and added new references to improve the text.

Lack of References in Discussion: Several statements in the discussion section lack proper referencing, making it challenging for readers to verify or contextualize the claims made.

Authors: All statements of the discussion were checked, and when necessary, new references were added.

Novelty and Hypothesis: The study does not clearly articulate its novelty in relation to existing literature. It's crucial to emphasize what sets this study apart from similar research. Additionally, the statement about successfully solving the hypothesis needs more robust evidence or explanation.

Authors: The initial hypothesis of our study was added to the new version of the manuscript, the novelty of our study was highlighted, and the relationship between our results and those of other studies was described in the Introduction and Discussion sections.

Ambiguity in Study Design and Methods: The study's methodology could be more detailed. For example, the specifics of the administration of adrenocorticotropic hormone (ACTH) are not mentioned, which is crucial for replication and understanding of the results.

Authors: As suggested, the description of methods used was checked and improved; for example, the details concerning ACTH administration were better described (lines 134-142).

Overall, while the study offers valuable insights into the effects of acute stressors on Saanen goats, addressing the above points would significantly strengthen its scientific rigor, clarity, and contribution to the field.

Authors: Thank you for your suggestions.

Round 2

Reviewer 2 Report

Comments and Suggestions for Authors

no comments